# Effects of High Intensity Dynamic Resistance Exercise and Whey Protein Supplements on Osteosarcopenia in Older Men with Low Bone and Muscle Mass. Final Results of the Randomized Controlled FrOST Study

**DOI:** 10.3390/nu12082341

**Published:** 2020-08-05

**Authors:** Wolfgang Kemmler, Matthias Kohl, Franz Jakob, Klaus Engelke, Simon von Stengel

**Affiliations:** 1Institute of Medical Physics, Friedrich-Alexander University of Erlangen-Nürnberg, Henkestrasse 91, 91053 Erlangen, Germany; klaus.engelke@imp.uni-erlangen.de (K.E.); simon.von.stengel@imp.uni-erlangen.de (S.v.S.); 2Faculty Medical and Life Sciences, University of Furtwangen, Neckarstrasse 1, 78054 Villingen-Schwenningen, Germany; Matthias.Kohl@hs-furtwangen.de; 3Bernhard-Heine-Center for Locomotion Research, University of Würzburg, Brettreichstrasse 11, 97074 Würzburg, Germany; f-jakob.klh@uni-wuerzburg.de; 4Department of Medicine III, Friedrich-Alexander University of Erlangen-Nürnberg, University Hospital Erlangen, Ulmenweg 18, 91054 Erlangen, Germany

**Keywords:** resistance exercise, osteopenia, sarcopenia, bone mineral density

## Abstract

The present study aimed to evaluate the effect of high intensity dynamic resistance exercise (HIT-DRT) and whey protein supplementation (WPS) on bone mineral density (BMD) and sarcopenia parameters in osteosarcopenic men. Men ≥ 72 years with osteosarcopenia (*n* = 43) were randomly assigned to a HIT-RT (HIT-RT: *n* = 21) or a non-training control group (*n* = 22). Supervised HIT-RT twice/week was applied for 18 months, while the control group maintained their habitual lifestyle. Supplying WPS, total protein intake amounted to 1.5–1.6 (HIT-RT) and 1.2 g/kg/body mass/d (control). Both groups were supplied with calcium and vitamin D. Primary study outcomes were BMD and the sarcopenia Z-score. After adjusting for multiplicity, we observed significant positive effects for sarcopenia Z-score (standardized mean difference (SMD): 1.40), BMD at lumbar spine (SMD: 0.72) and total hip (SMD: 0.72). In detail, effect sizes for skeletal muscle mass changes were very pronounced (1.97, *p* < 0.001), while effects for functional sarcopenia parameters were moderate (0.87, *p* = 0.008; handgrip strength) or low (0.39, *p* = 0.209; gait velocity). Apart from one man who reported short periods of temporary worsening of existing joint pain, no HIT-RT/WPS-related adverse effects or injuries were reported. We consider HIT-RT supported by whey protein supplementation as a feasible, attractive, safe and highly effective option to fight osteosarcopenia in older men.

## 1. Introduction

Osteopenia and sarcopenia are chronic conditions of advanced age with considerable pathophysiological overlap [1]. Apart from other health outcomes, both of these contribute significantly to the increase in the number of falls and fractures, corresponding hospitalizations, loss of independence, and higher mortality [2]. Although most studies focus on osteopenia and sarcopenia as separate conditions, there is emerging evidence that osteosarcopenia should be treated as a pathologic entity [3]. Muscle-bone interaction is extremely complex and both network with metabolism and the central nervous system in health and disease [4]. Hence, the clinical and largely geriatric syndrome of osteosarcopenia is also complex with respect to its pathophysiology that precipitates into different musculoskeletal (e.g., bone mineral density (BMD), muscle mass) and functional outcomes (falls, weakness, immobility). Aging and its associated conditions make the management of osteosarcopenia therapy a challenging mission. Dynamic-resistance exercise (DRT) supported by adequate dietary supplementation might be the most promising strategy for comprehensively improving the core important aspects of osteosarcopenia including metabolism, central nervous system (CNS), and cardiovascular endpoints [5] in older people. Indeed, to date, DRT and protein supplementation is the gold standard management for sarcopenia [6] and related risk factors e.g., falls and fall-induced fractures [7]. Further, low to moderate effect (size) of DRT on BMD has been reported [8]—albeit only for postmenopausal women. No exercise study on BMD has focused on male cohorts 70 years+, be it with or without sarcopenia or osteopenia. The few DRT studies with male cohorts [9,10,11,12], however, reported low to negligible effects on BMD. This unfavorable result can be attributed at least in part to suboptimum exercise programs or inadequate BMD assessments.

Thus, there is a need for innovative DRT trials on accepted endpoints of sarcopenia and osteopenia in older men. Considering the clinical relevance, we focused on a cohort of community-dwelling (cdw) men with osteosarcopenia.

Our primary hypothesis was that high-intensity, dynamic resistance exercise training (HIT-RT) supported by whey protein supplementation has a significantly positive effect (defined as difference from a non-training control group) on osteopenia and sarcopenia in older men with osteosarcopenia.

## 2. Materials and Methods

The FrOST (Franconian Osteopenia and Sarcopenia Trial) is an 18-month randomized parallel group exercise trial (RCT). The project focuses on the long-term effect of HIT-RT and whey protein supplementation on various risk factors of older cdw men with osteosarcopenia with special regard for muscle and bone parameters. The Institute of Medical Physics (IMP), University of Erlangen-Nürnberg (FAU), Germany initiated the project that was approved by the FAU ethics committee (number 67_15b and 4464b) and the federal bureau of radiation protection (BfS, number Z 5-2246212-2017-002). The project fully complies with the Helsinki Declaration [13]. All study participants gave their written informed consent after receiving detailed information. FrOST was fully registered under ClinicalTrials.gov: NCT03453463. The study was conducted between February 2018 (start of recruitment) and February 2020 (final data analysis). Two recent interim analyses after 6 and 8 months focus on parameters of sarcopenia [14], muscle mass and strength [15], while the 12-month FrOST analysis [16] addressed skeletal muscle mass index and BMD at the lumbar spine (LS) as determined by quantitative computed tomography (QCT).

### 2.1. Participants

Previous articles [14,15] reported the recruitment process of FrOST in detail, thus only a brief summary will be given here. In essence, 180 cdw men with low muscle mass [17], 72 years and older, living in the area of Erlangen-Nürnberg, Germany were willing to participate in the present study and were checked for eligibility. A further inclusion criterion was “osteosarcopenia,” i.e., (a) skeletal muscle mass index (SMI) ≤ 7.26 kg/m^2^ (i.e., lower than −2 SD T-Score: sarcopenia according to [18,19]) and (b) areal BMD T-Score lower than −1 SD at the lumbar spine (LS) or total hip (TH) region of interest (ROI (osteopenia/osteoporosis according to [20]). We excluded men with (a) secondary osteoporosis, (b) (osteo)anabolic and antiresorptive therapy with pharmaceutical agents, (c) glucocorticoid therapy > 7.5 mg/d for more than four weeks during the last 2 years, (d) hip fractures, (e) diseases, limitations or problems that prevent intense exercise or getting to the gym, (f) participation in any DRT during the last 2 years, (g) alcohol consumption > 60 g/d ethanol, (h) absence > 14 days during the study. Of the fifty eligible men with osteosarcopenia, seven refused to participate in the study predominately (*n* = 5) due to the inability to freely join the preferred study arm. Correspondingly, 43 men were allocated to a HIT-RT (*n* = 21) or control (CG, *n* = 22) group. Figure 1 shows the participant flow through the study.

### 2.2. Randomization Procedures

Using three strata (<6.70 kg/m^2^ versus 6.70–7.00 kg/m^2^ versus >7.00 kg/m^2^) for SMI, the men were randomly assigned to the two study arms. By drawing lots, participants allocated themselves to the HIT-RT or control group. Lots were placed in small opaque capsules (“kinder egg,” Ferrero, Italy) and freely drawn from a bowl by the participants. A researcher not involved in the present trial prepared the lots and supervised the procedure. Of importance, neither researchers nor participants knew the allocation beforehand (= allocation concealment). After randomization, participants were enrolled and instructed by the primary investigator (WK) about the dos and don’ts related to their study status.

### 2.3. Blinding

Due to personal relationships between the participants, we did not attempt to blind participants to their exercise status. However, we aimed to blind participants to the protein supplementation. Participants were informed that under consideration of their dietary protein intake both groups were equally provided with whey protein. Further, our blinding strategy focused on test assistants and outcome assessors who were kept unaware of the group status (HIT-RT or CG) of the participants and were not allowed to ask, either.

### 2.4. Study Procedure

FrOST aimed to determine the impact of HIT-DRT and WPS on LBM and BMD in an older cdw cohort of men with osteopenia and sarcopenia. However, we also supplemented cholecalciferol and calcium [21] (see below) according to present recommendations for both study groups. Whey protein doses of the HIT-RT were higher compared with the CG (1.5–1.6 g/kg vs. CG 1.2 g/kg body-mass/d; [22]) due to the increased HIT-RT induced protein demands. All participants were asked to maintain their dietary habits and lifestyle including physical activity and exercise routines outside the present study intervention.

### 2.5. Interventions

#### 2.5.1. Dietary Supplementation

As reported, HIT-RT and CG were provided with whey protein supplements (Active PRO80, inkospor, Roth, Germany) in order to generate a daily protein intake of 1.5–1.6 g/kg in the HIT-RT and 1.2 g/kg body-mass/d in the CG [22]. Individual supplementation with protein powder was based on 4-day dietary protocols (see below) completed by all participants at baseline, and then after 28, 54, and 78 weeks. The protein product had a chemical score of 159; 100 g protein powder (360 kcal) contained 80 g of (whey) protein. Essential amino acid amount was 57 g; 100 g of the product contained 9 g of L-Leucine. Of importance, 1200 mg of calcium were contained in 100 g of the protein product. We recommended participants mix the protein powder with low fat milk (10 g/150 mL) and to split doses of more than 40 g (*n* = 3). A specific time of the day to consume the protein was not specified [23]. All participants received detailed written information and instructions on protein intake.

#### 2.5.2. Vitamin D and Calcium Supplementation

Based on blood sampling and analysis of 25-Hydroxy-Vitamin-D (25-OHD) at baseline and after 54 weeks, all participants were provided with cholecalciferol (MYVITAMINS, Manchester, UK). Participants with serum concentrations below 30 ng/mL (*n* = 37) were supplemented with 10,000 IE/week (4 units × 2500 IE). Participants with serum levels of between 30–40 ng/mL (*n* = 6) were requested to take two capsules of 2500 IU throughout, i.e., 5000 IE/week.

Considering the calcium provided by the protein product, calcium questionnaires (Rheumaliga, Switzerland) conducted at baseline, and then after 28, 54, and 78 weeks were used to determine the calcium intake of the participants. We intended to realize a total calcium intake of ~1000 mg/d [21] in all the men. Calcium capsules (Sankt Bernhard, Bad Dietzenbach, Germany) that contained 250 mg of pure calcium were provided for participants below 950 mg/d of dietary intake. Of importance, capsules can easily be halved to generate a lower dose.

#### 2.5.3. Resistance Exercise

The 18-month FrOST intervention started in June 2018 with an emphasis on a high-intensity resistance exercise training (HIT-RT). We focused exclusively on isolated DRT on machines, i.e., we did not conduct any other types of exercises either in parallel or as a warm-up before the HIT-RT session. All the exercise sessions took place in a centrally located gym (e.g., Kieser Training, Erlangen, Germany) on Mondays, Wednesdays, and Fridays between 8:00 and 10:00 a.m. Two training sessions per week were scheduled; however, participants were allowed to exercise three times before or after week(s) of temporary inability (holidays, illness). A licensed instructor consistently supervised all exercise sessions. Our training strategy concentrated on short but intense bouts of exercise. Consequently, after a careful conditioning period, we started the periodized HIT-RT approach defined as single set exercise training with high intensity and effort using intensifying strategies [24]. All major and most of the minor muscle groups were exercised by a progressively increased maximum of 12–14 exercises/session. Exercises were selected from a pool of 18 exercises (calf raises, leg press, -extension, leg curls, -adduction, -abduction, hip extension, latissimus front pulleys, pull-overs, seated rowing, back extension, inverse fly, bench press, military press, lateral raises, butterfly with extended arms, crunches, lateral crunches). We worked with detailed training logs that prescribed exercises, repetitions (reps), movement velocity/duration, and “effort”. “Effort” (or absolute intensity) was prescribed by a range of reps (i.e., 5–7 or 8–10 reps) and the corresponding degree of work to failure (e.g., maximum effort minus 1–3 reps; defined as non-Repetition Maximum (nRM [25]): “Set endpoint when trainees complete a pre-determined number of repetitions despite the fact that further repetitions could be completed.”) [26,27]. We specified set endpoints in “non-repetition maximum” (nRM) and “(self-determined) repetition maximum” (….however, apart from a short testing phase, we abstain to prescribe “complete momentary muscular failure”(MF) [25] during the intervention) (RM) [25].

The 18-month intervention was structured into periods of 8–12 weeks, each with dedicated training protocols and a progressive increase in exercise intensity during the intervention. Due to the rather complex HIT-RT protocol and to motivate participants to work with high compliance, joint briefing meetings were held to explain and discuss the next training phase.

Phase 1. The first four weeks of the intervention were used for familiarization with the exercise protocol and learning of lifting techniques. A further eight weeks of conditioning concentrated on briefing and experience to select the adequate load to the prescribed varying repetitions. Sets of 8–15 reps, using nearly the full range of motion were prescribed. Time under tension (TUT)/rep was prescribed in a 2 s concentric, 1 s isometric, and 2 s eccentric (2 s-1 s-2 s) phase with 90 to 120 s of rest pauses. Incomplete work to failure [25] was scheduled during this introductory training phase.

Phase 2. Two linearly periodized 4-week phases, with each fourth week with low effort (i.e., recovery week), were applied during the second study period. Participants had to select a load that ensured a repetition maximum −1 rep (5–10 reps) or −2 reps (10–18 reps) [28]. Movement velocity varied between the sessions from TUT: 4s – 1s – 4s/rep to 1s –to 1s – 2s/rep.

Phase 3. We introduced explosive movements during the concentric phase and the repetition maximum (RM) (“Set endpoint when trainees properly complete the final repetition possible but if the next repetition were attempted they would definitely achieve momentary failure” [25]. In contrast, sets/exercises with high velocity had to be stopped when explosive movement in the concentric phase failed) approach [25], the latter only for sets ≤10 reps. Apart from some exercises (i.e., back extension), 25%–35% of the sets of a session were prescribed with an explosive movement. Of note, sets conducted with explosive movements were consistently nRM sets. Ninety seconds (after nRM sets) – 120 s (after RM sets) of rest between the sets was applied.

Phase 4: Supersets that included sets either for the same or related muscle groups (e.g., bench press and butterfly), or agonist and antagonist (e.g., leg extension and leg curls) in a row. Two or three exercises were included in a superset sequence. After a superset, 30–45 s of rest within a superset block and two minutes of rest was prescribed.

Phase 5: The drop-sets approach was introduced: i.e., participants work to RM (≤10 reps) or RM −1 rep (>10 reps), decrease the load by 10%–20% and, without rest, immediately work again to RM. Seven supersets exercises were addressed by the drop-set approach. One minute of rest within and 2 min between the supersets were prescribed.

Phase 6: We introduced the momentary failure approach (…briefly “inability to realize the concentric phase of the current rep”) after an exhaustive discussion of this method in the “joint briefing and information session.” However, after the first weeks, participants expressed a lack of willingness for this rather exhausting protocol, and so we switched back to the periodized superset/drop-set approach to RM, with varying movement velocity and number of reps applied in the previous phase. The only novelty of phase 6 was that the load was deceased twice within the drop set of the moderate and high effort week (i.e., second and third week of the 4-week cycle).

Phase 7 and 8: During the last 16 weeks of HIT-RT, we did not make any significant changes to the protocol (compared to phase 6). Slight variations of the exercise order within or between the supersets and a more pronounced variation of the number of reps (4–6 to 16–20 reps) were introduced.

For more details of the HIT-DRT protocol the reader is kindly referred to previous FrOST publications.

#### 2.5.4. Compliance with the Intervention

Attendance rate and duration of the session were accurately determined by the gym’s chip card system. In parallel, participants’ training logs were reviewed after each training phase to determine participant compliance with the exercise protocol (particularly for “effort”) by considering the relation of prescribed number of repetition and applied load as listed by the participants. This estimation was based on predicting equations that estimate 1RM from repetitions to fatigue (and vice versa) [29] for all exercises except back extension. Data were consistently collected during a 6–10 rep RM sessions with a movement velocity of 2 s-1 s-2 s (up from phase 3). In this range of TUT (4–7 s) we defined a difference of ≥10% between load selected by the participant for a given number of reps under RM and calculated load as inadequate. In parallel, during sessions instructors particularly checked participants’ compliance to exercise with proper intensity/effort.

Participants’ adherence to the protein, Vit-D, and calcium supplementation was checked by three independent strategies. (a) We checked our distribution logs for protein, Vit-D, and calcium; (b) participants were phoned biweekly by a research fellow; (c) we carried out personal interviews at all follow-up (FU) assessments.

### 2.6. Study Outcomes

Due to our hypothesis of DRT-effects on “osteo-sarcopenia,” we had to address outcomes for sarcopenia and for osteopenia/osteoporosis. However, considering the main fracture sites [30], we applied two outcomes to address osteoporosis: areal BMD at the lumbar spine (LS) and areal BMD at the total hip (TH). In order to generate one continuous rather than a series of dichotomous scores to determine changes of parameters constituting sarcopenia according to the European Working Group on Sarcopenia in Older People (EWGSOP) I criteria (..the study started before the EWGSOP II definition of Sarcopenia was published [31]), we used a Z-Score as applied in previous studies [32,33] to summarize the morphometric and functional aspects of sarcopenia (i.e., SMI, handgrip strength, and gait velocity; [19]).

#### 2.6.1. Primary Study Outcomes


Sarcopenia Z-Score according to EWGSOP I [19] at baseline and immediately post-intervention (18 months).Areal bone mineral density (aBMD) at the Lumbar Spine (LS) as determined by Dual Energy X-ray absorptiometry (DXA) at baseline and immediately post-intervention.Areal bone mineral density (aBMD) at the total hip as determined by DXA at baseline and immediately post-intervention.


#### 2.6.2. Secondary (i.e., Explanatory) Study Outcomes Related to Sarcopenia


Skeletal Muscle Mass Index (SMI) as determined by Dual-Energy Absorptiometry (DXA) at baseline and immediately post-intervention.Handgrip strength at baseline and immediately post-intervention.Gait velocity at baseline and immediately post-intervention.


#### 2.6.3. Changes in Trial Outcomes after Trial Commencement

No changes of trial outcomes were made after the start of the trial. However, due to a technical failure of the DXA-Scanner, body composition was determined after 36 weeks instead of 28 weeks as intended.

### 2.7. Assessments

The final assessments took place after the regeneration week of the last mesocycle. We placed great emphasis on standardizing assessments. The same research assistant consistently led, supervised, and analyzed the assessment in question at about the same time of day (±2 h). Tests were consistently conducted at the same location, in the identical order and with the same calibrated devices. Further, we requested participants to maintain their physical activities and diet and to avoid intense physical activity 48 h prior to the tests.

Body height was assessed by a Holtain stadiometer (Crymych Dyfed, Great Britain) and body mass was determined by the scale function of a direct-segmental, multi-frequency Bio-Impedance-Analysis (DSM-BIA; InBody 770, Seoul, Korea) (corresponding data on body composition as assessed by BIA technique will be not reported here). Body composition, (areal) BMD at the lumbar spine, TH, and total body ROI were evaluated using a DXA-Scanner (QDR 4500a, Discovery-upgrade, Hologic Inc., Bedford, MA, USA). All scans and analyses were applied according to the manufacturer’s specifications. Using the total body scan specification, amongst other parameters, appendicular skeletal muscle mass (ASMM, i.e., fat free mass of the lower and upper extremities) was segmented using the “compare mode” at all follow-ups. Correspondingly, area and placement of the baseline specifications were exactly reproducible. Based on ASMM, skeletal muscle mass index was defined and calculated as ASMM/body-height (kg/m^2^) [18]. Long-term coefficient of variation for aBMD at the LS from study start to 78-weeks assessment was 0.44%.

We applied the habitual gait speed protocol recommended by Kressig [34] using the 10 m protocol recommended for research [35]. Participants started walking in an upright position 3 m before the first photosensor (HL 2–31, TagHeuer, La Chaux-de-Fonds, Switzerland) and stopped 2 m after the second photosensor, allowing time to accelerate and decelerate. We asked men to wear regular shoes and performed tests without any specific walking aids. Our standardized instruction to the participants was consistently: “walk at a speed just as if you were walking along the street to go to the shops.”

A calibrated Jamar handgrip dynamometer (Sammons Preston Inc., Bollington, UK) was used to assess handgrip strength. We adjusted handgrip width of the device individually to participant hand size. Tests were performed three times on the dominant and the non-dominant hand while standing in an upright position with arms down by the side [36]. The standardized instruction to the participants was consistently: “squeeze as strongly as possible.” The highest result of the three trials for the dominant hand was included in the analysis.

We summarized sarcopenia parameters to a joint score in order to avoid a further complex multiple test problem. Applying the EWGSOP-I definition of sarcopenia which includes SMI, gait velocity, and handgrip strength, and applying the corresponding cut-off values i.e., 0.8 m/s for gait velocity, 30 kg for handgrip strength, and 7.26 kg/m^2^ for SMI, we calculated the sarcopenia Z-Score as:Z-Score = ((30 − individual handgrip strength)/SD handgrip strength) + ((0.8 − individual gait velocity)/SD gait velocity) + ((7.26 − individual SMI)/SD SMI).(1)

At baseline, participants’ (a) demographic parameters, (b) diseases, limitations, operations, (c) pharmacologic therapy, dietary supplements, (d) number of falls and injurious falls within the last year, (e) injuries and low trauma fractures within the last year, (f) lifestyle, including physical activity and exercise, and (g) independence status were recorded using a standardized questionnaire [37,38]. Follow-up questionnaires that focused particularly on changes during the study period which might affect our study result were conducted after 28, 36, 54, and 78 weeks. Here, we placed high emphasis on potential changes of physical activity and exercise habits and [38], and in particular on changes of medication and supplements, and emerging or worsening of existing complaints or diseases. In order to generate reliable and complete records, we required participants to list their medications, supplements, limitations and diseases at home and to provide the documents for the follow-up assessments. Here, documents and completed questionnaires were lastly checked for consistency, completeness, and reasonability in close interaction between the primary investigator and participants.

To adequately realize and monitor our intended protein intake of 1.5–1.6 g/kg/d in the HIT-RT and 1.2 g/kg/d body mass in the CG, all the participants took four-day diet records at baseline and also after 28, 54, and 78 weeks. We provided all participants with standardized diet records (Freiburger Nutrition Record, nutri-science, Hausach, Germany). Briefly, the Freiburger Nutrition Record is a tally list of how often specified food products were consumed. Based on this list the corresponding software calculated energy and macronutrients intake. After being carefully instructed on how to keep the records, participants recorded 3 weekdays and one weekend day considered characteristic for their nutritional habits. The same research assistant consistently analyzed all the diet records. In cases of unrealistic results (e.g., protein intake <50 g or >150 g/d) or relevant individual changes between the assessments, the results were discussed with the participants, and in all cases another diet record based on more representative days had to be completed.

### 2.8. Sample Size Analysis

Considering that resistance-type exercise combined with whey protein supplementation has been reported to be highly effective in addressing sarcopenia in older men [33], our sample size calculation concentrated on the more critical aspect of osteosarcopenia, i.e., osteopenia/osteoporosis. Thus, our power calculation was based on areal BMD at the lumbar spine (LS) and total hip (TH). Assuming an exercise effect (Δ-HIT-RT vs. Δ-CG) on BMD-LS and BMD-TH of 2.0 ± 2.0% and adjusting for multiplicity (LS-BMD, TH-BMD, SMI) using the Bonferroni-Holm method, [39] 21 participants per group were required to detect a type-I error of alpha = 0.024 with 80% power (1-β) (t-test based sample size calculation).

### 2.9. Statistical Analysis

We applied intention-to-treat (ITT) analysis that included all participants who were randomly assigned to the study arms (HIT-RT vs. CG) regardless of their loss to follow-up, compliance, or confounding aspects. We applied multiple imputation (ITT) using R statistics software (R Development Core Team Vienna, Austria) in combination with Amelia II [40]. We used the full data set for multiple imputations, and repeated imputation 100 times. As confirmed by over-imputation diagnostic plots (“observed versus imputed values”) provided by Amelia II, imputation worked well in all cases. We checked normal distribution of the data by statistical (Shapiro-Wilks) and graphical (qq-plots) tests. All the study outcomes addressed here were analyzed by dependent *t*-tests. To identify differences between changes in the HIT-RT and CG, we applied t-test comparisons with pooled SD. All tests were 2-tailed and significance was accepted at *p* < 0.05. Effect sizes were indicated by standardized mean difference (SMD) according to Cohen (d’ [41]). In order to adjust for multiple testing, a problem inherent in the present research question on “osteosarcopenia,” we applied the Bonferroni-Holm method [39]. Of note, we did not adjust secondary (explanatory) study endpoints for multiple testing [42]. All statistical procedures were conducted by a professional statistician (MK).

## 3. Results

### 3.1. Participant and Exercise Characteristics

Table 1 shows participant baseline characteristics. Apart from baseline protein intake that was remarkably high in the CG, no significant differences were observed between the HIT-RT and control group. However, there were some noticeable features. Apart from a very low baseline 25-OHD levels in both groups, applying an obesity cut-off of 27%–30% total body fat, as suggested by the majority of SO-studies [43], found that the vast majority (75%–94%) of our cohort was “osteosarcopenic obese” (SO) [44].

Three men of the CG and two men of the HIT-RT were lost to follow-up (Figure 1). Two participants lost interest, one participant of the HIT-RT suffered from prostate cancer and withdrew from the study after 3 months, and two other CG participants were unable to attend the 78-week assessment due to a hip fracture or an influenza infection.

The attendance rate was very high (95% ± 5%). Thirteen men completed all of their prescribed 156 sessions; only one man who reported temporary knee and shoulder problems conducted fewer than 125 sessions (80%). As determined by the training logs and confirmed by the trainers, most specifications of the training protocol were properly met. However, exercise intensity remains an ongoing problem. Applying our approach of 10% difference between individually selected load and calculated load, 25%–35% of the corresponding RM sets were performed with too inadequate effort. During the regular exercise sessions, we did not observe any unintended side effects or injuries. However, one man reported temporary worsening of existing knee and shoulder pain during and after the exercise sessions.

Monitored by our biweekly phone calls, supply logs, and personal interviews at 28-, 36-, 54-, and 78-week FU-assessments, we rate compliance with the protein, calcium, and Vit-D supplementation as high. In detail, all men reported they adhered exactly to the specifications. This statement was consistent with our supply logs.

### 3.2. Primary Study Outcomes

The sarcopenia Z-Score summarizing SMI, grip strength, and gait velocity according to EWGSOP I [19] improved significantly in the HIT-RT (*p* < 0.001) and worsened significantly (*p* = 0.018) in the CG (Table 2). Changes between the groups significantly (*p* < 0.001) differed; SMD (1.40) indicated high effect size.

LS-BMD was maintained in the CG (0.0 ± 1.6%; *p* = 0.807) and increased significantly in the HIT-RT (0.9 ± 1.3%; *p* = 0.006). In summary, a significant difference in BMD changes of the LS was determined between the groups (*p* = 0.0237; SMD = 0.72) (Table 2).

TH-BMD decreased significantly in the CG (−1.6 ± 2.1%; *p* = 0.003) and was maintained in the HIT-RT (0.0 ± 1.3%, *p* = 0.847). Changes in TH-BMD differed significantly between the groups (*p* = 0.0247; SMD = 0.72) (Table 2).

After adjusting for multiple testing (SMI, LS-, TH-BMD) using the Bonferroni-Holm method [39], effects remained significant (Sarcopenia Z-Score: *p* < 0.001; LS-BMD: 0.047; TH-BMD: 0.047).

### 3.3. Secondary Study Outcomes

Table 3 displays baseline values and changes in secondary study outcomes related to sarcopenia. In summary, the HIT-RT demonstrated a significant increase in SMI (*p* < 0.001) and handgrip strength (*p* = 0.003), but not for gait velocity (*p* = 0.794), while SMI (*p* = 0.003), gait velocity (*p* = 0.039), and handgrip strength (*p* = 0.429) changed negatively in the CG. Effects sizes vary from SMD d’ = 1.97 for SMI to d’ = 0.39 for gait velocity (Table 3).

### 3.4. Confounding Parameters

No substantial changes (all *p*-values > 0.576) in dietary intake parameters (e.g., energy, macro-nutrition) from baseline were observed at 28, 54, and 78-week FU. Calcium intake as determined by the calcium questionnaire provided by the Rheumaliga Suisse was stable in the EG and CG (*p* > 0.444) during the interventional period and did not differ much at baseline, 28, 54, and 78-week FU. Of note, due to the calcium included in the protein powder mixed with low fat milk (if applicable), and the higher protein doses provided in the HIT-RT, only two participants of the HIT-RT were supplemented with calcium capsules. On the other hand, 14 participants of the CG required supplements to realize the recommended intake of 1000 mg/d. Dietary protein intake according to the dietary protocols significantly varied between the groups at baseline (Table 1) and at 54-week FU. However, changes within the HIT-RT or CG were non-significant (*p* > 0.601; 0.05–0.11 g/kg/d) on average. Due to the rather high dietary protein intake of the CG (Table 1), only 30% of the corresponding participants were supplemented with whey protein powder (versus 95% in the HIT-RT). With respect to total protein intake as determined by the dietary protocol at 28, 54, 78-week FU and our records and dietary logs (…verified by personal interviews), protein intake varied between 1.21 ± 0.13 and 1.24 ± 0.11 g/kg/d in the CG and 1.56 ± 0.14 to 1.61 ± 0.16 g/kg/d body mass in the HIT-RT. Average difference in total whey protein consumption varied between 0.34 (12 months) and 0.39 g/kg/d (6 months) between CG and HIT-RT.

Cholecalciferol supplementation was grounded on baseline (April 2018) and 54-week assessments of 25-OHD levels. At baseline, 11 participants each of the HIT-RT and CG demonstrated serum concentrations of 25-OHD below the present recommendations [21]. Due to the cholecalciferol supplementation, 25-OHD levels increased significantly from baseline (Table 1) to 54-week FU (HIT-RT: 25.2 ± 5.6 vs. CG: 26.8 ± 7.9), Nevertheless, on average 25-OHD levels still ranged below present recommendations. Although we did not change our cholecalciferol supplementation, 25-OHD increased slightly by 78-week FU (HIT-RT: 28.1 ± 6.1 vs. CG: 29.6 ± 5.8). However, of note, about half of our participants showed 25-OHD levels below 30 ng/mL “throughout the study” (…or at least at baseline, after 54 and 78 weeks).

Physical activity and exercise habits did not change noticeably (*p* < 0.501) or vary between the groups (*p* < 0.357). Apart from the man who suffered from prostate cancer, two men of the HIT-RT reported training breaks (of four weeks and two months) due to a respiratory tract infection or worsening of knee/shoulder pain. In parallel, three men of the CG reported periods of physical inactivity (3–6 weeks) caused by diseases or surgery/hospitalization.

## 4. Discussion

From an interventional and biometrical point of view, addressing “osteosarcopenia” [46], “sarcopenic obesity” [47] or even “osteosarcopenic obesity” [44], with their inherent combination of at least three different conditions, is a daunting task. In general, exercise or more dedicated resistance exercise—ideally supported by dietary supplements—might be the most promising strategy to positively affect all physiologic and functional outcomes related to the osteopenia/sarcopenia/obesity complex [48,49]. However, due to the gynocentrism of osteoporosis research, there is a lack of exercise studies [50] confirming the positive effects on BMD demonstrated for female cohorts [8,51] in older men. Considering further that muscles and bones interact intensely [4] and are regarded as “neighbors with close relationships” [52] or even “Siamese twins” [53], it appears reasonable to address both conditions, osteopenia and sarcopenia, in men with a combined DRT protocol.

In the recent 12-month publication of the present FrOST project [16], we reported positive effects on SMI and integral volumetric BMD at the LS as determined by quantitative computed tomography (QCT); however, we did not determine significant effects on areal BMD at LS and total hip as determined by the less sensitive but nevertheless gold standard DXA-technique. The present (final) analysis after 18 months of progressive DRT added evidence that HIT-DRT supported by WPS (1) significantly improved the sarcopenia complex composed of morphometric (SMI) and functional (handgrip-strength, gait velocity) [19] aspects and (2) areal BMD at the most important fracture sites, i.e., LS and total hip. Thus, our results clearly indicate the favorable effect of a time-efficient, high-velocity/high-intensity/high-effort DRT combined with moderate dosed WPS on osteosarcopenia, as determined by LS- and TH-BMD and sarcopenia Z-Score in this cohort of cdw 73–91-year-old men with osteosarcopenia. Based on previous studies [32,33], we anticipated a pronounced effect on muscle mass and function, whilst estimating the DRT effect on BMD would be much less pronounced [50]. Correspondingly, our DRT protocol applied a periodized, high-intensity/high-effort/high-velocity HIT-RT protocol dedicated to bone [54]. Key features of our DRT protocol were: (a) application of axial loading of the LS and/or hip ROI by selected exercises (e.g., (semi-supine) leg press, calf raises, military press); (b) high strain magnitude [55] applied though high loads; (c) high strain rates [56] applied by fast movement velocity (i.e., “power”); [57] (d) varying strain duration/rep by varying time under tension (TUT); (e) varying strain frequency (0.1 to 0.5 Hz) [58] through varying TUT; and (f) aspects related to bone desensitization [59] via the periodized approach. Further, we applied short but intense bouts of exercises known to generate a favorable anabolic hormonal milieu (review in Schroeder et al. [60]) to increase the sensitivity of bone cells to mechanical loading [61,62]. However, beyond bone, the same DRT protocol generated high muscular tension, muscle damage, and metabolic stress, i.e., key aspects of the hypertrophic response to resistance exercise [63]. Considering further that the protocol includes sequences of high movement velocity (i.e., power), we feel that not only muscle strength but also lower extremity muscle function was addressed to an appropriate degree [64].

In summary, we found a very pronounced hypertrophic effect (i.e. about 1.50 kg on soft fat free mass) that considerably exceeded corresponding data on fat-free mass reported by the present literature, be it with or without supplements (e.g., protein, Vit-D, creatine) [65,66,67,68,69,70]. Although we observed a leveling-off effect of SMI (…based on BIA assessments, 85% of SMI increase in the HIT-RT occurred within the first 28 weeks [14]. This finding indicates that our higher effect on SMI was not based predominately on the longer study duration, however), it nevertheless improved up to the final, i.e., 18-month, FU. Surprisingly, less impressive findings were demonstrated for grip strength and particularly for gait velocity, i.e., functional aspects of sarcopenia according to EWGSOP-I. However, one should be aware that handgrip strength is not the most sensitive measure of muscle strength changes during resistance training [71]. Indeed, the highly significant effect on maximum hip-/leg-extensor strength as determined by an isokinetic leg press after one year (~30 ± 15%) [16] might confirm the low sensitivity of handgrip strength for resistance training-induced strength changes. This might also be the case for gait velocity. Even though gait velocity is considered to be a proxy for lower extremity muscle function [19], lower extremity muscle function is not the only factor impacting gait speed. Age-related motor neuron degradation [72], range of motion of the lower extremities joints [73], and non-muscular factors (e.g., cognitive status [74] and depression [75]) impact gait speed, while muscle mass plays a minor role [72]. In parallel, the absence of positive results on gait velocity in our study might be further related to the high baseline gait velocity and a corresponding ceiling effect.

Revisiting BMD, we observed significant effects at the LS and TH that in essence remained significant after adjusting for multiple testing. Due to a lack of corresponding exercise studies, it is difficult to compare our finding with the present literature. Applying a combined high impact weight-bearing and DRT protocol in men 50–79 years old, Kukuljan et al. [76] were the only researchers to determine significant effects on BMD in men, albeit only at the hip-ROI.

Apart from the “osteosarcopenia” aspect, another novelty of our research on older people was the HIT-RT approach. However, one should consider that our protocol was not a completely “purebred” HIT-RT. The biggest difference was that, apart from a short “try out” phase, we did not prescribe work to failure (MF), which is characteristic for HIT-RT protocols [24]. Correspondingly, our approach of progressively increasing the general intensity of the exercise protocol was centered on intensifying strategies (e.g., supersets or drop sets). Further, since sets with explosive movements were not conducted with RM, and other exercises (i.e., back extension) or sets were prescribed with nRM, even in high effort weeks (i.e., each third week of a four-week mesocycle), the amount of RM sets never exceeded 60% of volume/session. Another, less intended but nevertheless expected, aspect was that up to 30% of the sets scheduled to RM were conducted with lower intensity. We consistently briefed and encouraged participants accordingly, however, we (have to) accept that we were not working with athletes willing to consistently go to their limits.

Apart from the “high effort issue” and high relative intensity (up to 85% 1RM), we prescribed movements with explosive velocity. One may argue that this approach might be inappropriate for older men with osteosarcopenia; however, we [77] and others [78,79] did not observe any relevant adverse effects of high intensity (%1RM)/high velocity, at least when carefully introduced, consistently supervised, and conducted on RT-machines. This observation is confirmed by the present study not leading to any exercise-induced injuries or orthopedic complaints other than delayed onset muscular soreness, apart from one man who reported temporary worsening of existing knee and shoulder pain. Nevertheless, the combination of high intense exercise and increased protein doses might lead to negative renal (or cardiac) consequences [80,81]. So far, we have not addressed this issue in FrOST, a recent study we conducted on whole-body electromyostimulation (WB-EMS) and protein supplementation (1.7–1.8 g/kg/body mass/d) generated no negative effects in a similar cohort of men with sarcopenic obesity [82]. Thus, in summary we consider our intervention to be safe and applicable in this cohort of older men with musculoskeletal limitations.

Another feature of the intervention was the supplementation of whey protein. We generated a similar total Vit-D and Ca-intake in the HIT-RT and CG according to current guidelines [21], however, according to recent recommendations [22], we intended a higher total protein intake in the HIT-RT (1.5–1.6 vs. CG: 1.2 g/kg/d body mass) to account for increased muscular repair and adaptation processes. Thus, our protein approach focused predominately on the muscular aspect of osteosarcopenia; but there is some evidence for a significant superior effect of high versus low protein intake on BMD at the LS [81,83]. On the other hand, the additional effect of protein supplementation to resistance exercise on muscle mass and/or function in older people with [49,84,85] or without sarcopenia/SO [69,85] is still under discussion. In general, FrOST did not address the research issue of different contributions of HIT-RT vs. whey protein supplementation on osteosarcopenia. However, the significantly higher baseline protein intake of the CG (Table 1) and the corresponding difference in whey protein powder supplementation between the groups might partially explain the exceptionally high study effects on SMI (or LBM). Thus, although we are unable to assess the dedicated impact of HIT-RT vs. protein supplements, we speculate that the whey protein supplementation might have contributed considerably to our findings, be it for muscle or bone mass.

Some features and limitations of FrOST should be noted to adequately estimate the study findings and conclusions. (1) Our trial focuses on men. This decision was based on (a) the lack of corresponding exercise studies in the otherwise gynocentric area of osteoporosis; (b) the aspect that no DRT study in men reported significant effects on BMD, and (c) considering that comparable exercise protocols were applied, there might be some evidence that exercise-induced effects on BMD differ between the genders. (2) Our corresponding inclusion criteria focus on SMI, i.e., the morphologic aspect of sarcopenia; functional aspects were not considered. (3) We summarized functional and morphometric components of sarcopenia using a comprehensive Z-Score [32,33] in order to reduce the degree of multiplicity inherent when addressing ”osteo-sarcopenia” as a study outcome. Due to the study start in 2018, we applied cut-off values suggested by the recently revised EWGSOP-I definition [19]. (4) Regarding “osteopenia,” we included both LS-BMD and TH-BMD as main fracture sites [30], which was not in line with our strategy to focus on as few (primary) outcomes as possible to reduce the degree of multiplicity. However, the complete lack of exercise studies on BMD in older men with osteoporosis may justify this approach. (5) Due to the multiplicity problem coupled with the limited number of men with osteosarcopenia, we decided to apply the more powerful method of Bonferroni-Holm [39] in sample size analysis and further statistical procedures. (6) We applied a time-efficient HIT-RT twice a week, potentially attractive for (older) people unmotivated or unable to exercise more frequently [86]. Indeed, the present attendance and low drop-out rate supports the high attractiveness and feasibility of the exercise program. (7) We observed no injuries and orthopedic complaints, however, considering the potentially negative effect of HIT-RT and protein supplementation on renal hepatic and cardiac function, this issue should be addressed in the near future.

## 5. Conclusions

In conclusion, applying careful conditioning, consistent supervision, and working only to RM, HIT-DRT on devices combined with moderate protein supplementation is a safe, attractive, and highly effective and efficient approach for addressing osteosarcopenia in older cdw men with sarcopenia and osteoporosis. So far, FrOST might serve as a blueprint for an exercise protocol applicable in older men with low bone mass, muscle mass, and muscle function. Bearing in mind that sarcopenia and sarcopenic obesity are related to inflammation [87], mitochondrial abnormalities, [88] and oxidative stress [89] (applying a reasonable cut-off for (sarcopenic) obesity (i.e., 27%–30% bodyfat [43]), about 75%–90% of our cohort featured sarcopenic obesity), we speculate that changes in bone and muscle mass might be even more pronounced in healthy older people.

## Figures and Tables

**Figure 1 nutrients-12-02341-f001:**
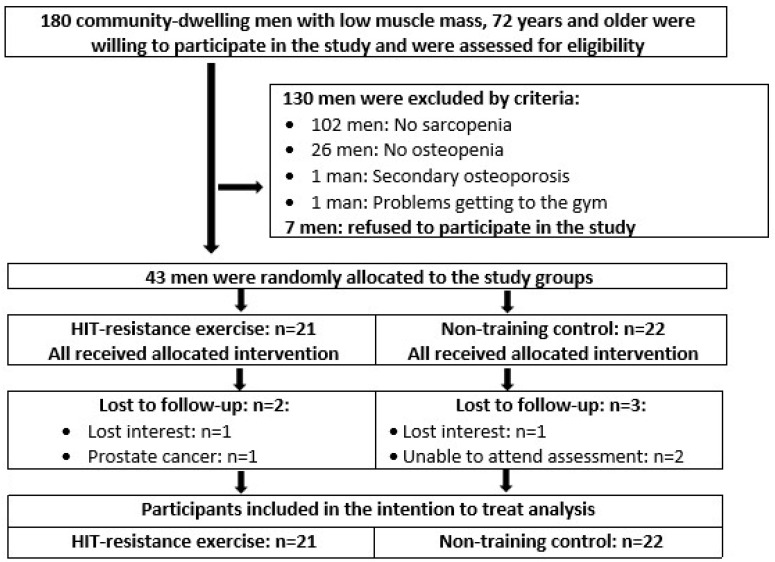
Participant flow through the study. HIT: High intensity training.

**Table 1 nutrients-12-02341-t001:** Baseline characteristics of the participants of high intensity resistance exercise training group (HIT-RT) versus the control groups (CG). MV ± SD: mean value ± standard deviation.

Variable	CG (*n* = 22) MV ± SD	HIT-RT (*n* = 21) MV ± SD	*p*
Age [years]	79.2 ± 4.7	77.8 ± 3.6	0.262
Body Mass Index (kg/m^2^)	24.5 ± 1.9	25.0 ± 3.0	0.515
Total Body Fat (DXA) (%)	33.6 ± 4.0	34.5 ± 6.1	0.563
Osteoporosis (*n*) ^a^	8	7	0.757
More than two diseases (*n*) ^b^	12	10	0.826
Lower limb arthritis (*n*) ^b^	2	2	0.959
Diabetes Mellitus type II (*n*)	1	1	0.960
Habitual gait velocity (m/s)	1.26 ± 0.15	1.25 ± 0.17	0.703
Handgrip strength (kg)	30.0 ± 4.3	30.7 ± 5.1	0.675
Physical activity (Index)^c^	4.15 ± 1.53	4.45 ± 1.32	0.490
Exercise ≥1x week (*n*)	5	5	0.931
25-OHD (ng/mL) ^d^	17.5 ± 7.0	21.6 ± 8.4	0.126
Calcium intake (mg/d) ^e^	833 ± 282	802 ± 226	0.636
Energy intake (kcal/d) ^f^	2291 ± 590	2155 ± 416	0.407
Protein intake (g/kg/d) ^f^	1.29 ± 0.34	1.10 ± 0.25	0.043

^a^ <−2.5 SD-T-Score at LS or total hip; ^b^ ICD-10 based disease cluster of Schäfer et al. [45]; ^c^ scale from (1) “very low” to (7) “very high” [38]; ^d^ Roche Diagnostics, Mannheim, Germany; ^e^ as determined by a Calcium Questionnaire provided by Rheumaliga, Switzerland); ^f^ as determined by dietary records.

**Table 2 nutrients-12-02341-t002:** Baseline data and changes in primary study endpoints in the high intensity resistance exercise training group (HIT-RT) versus the control groups (CG). MV (95%-CI): mean value (95%-confidence interval). * unadjusted *p*-values.

	CG MV (95% CI)	HIT-RT MV (95% CI)	Difference MV (95% CI)	*p*-Value
**Sarcopenia Z-Score**				
Baseline	−2.14 (−1.45 to −2.83)	−2.51 (−1.45 to −3.65)	-------------	0.558
Changes	0.48 (0.13 to 0.82)	−0.83 (−0.49 to −1.17)	1.31 (0.74 to 1.89)	<0.001 *
**Areal bone mineral density at the lumbar spine (mg/cm^2^)**				
Baseline	0.987 (0.916 to 1.060)	1.054 (0.981 to 1.122)	------------	0.140
Changes	−0.001 (−0.008 to 0.005)	0.011 (0.004 to 0.017)	0.012 (0.001 to −0.020)	0.024 *
**Areal bone mineral density at the total hip (mg/cm^2^)**				
Baseline	0.869 (0.826 to 0.911)	0.894 (0.856 to 0.932)	------------	0.364
Changes	−0.013 (−0.021 to −0.007)	−0.000 (−0.008 to 0.006)	0.013 (0.002 to 0.022)	0.025 *

**Table 3 nutrients-12-02341-t003:** Baseline data and changes in secondary (explanatory) study endpoints in the high intensity resistance exercise training group (HIT-RT) versus the control groups (CG). MV (95%-CI): mean value (95%-confidence interval). * unadjusted *p*-values.

	CG MV (95% CI)	HIT-RT MV (95% CI)	Difference MV (95% CI)	*p*-Value
**Skeletal Muscle Mass Index (SMI) (kg/m^2^)**				
Baseline	6.89 (6.74 to 7.02)	7.01 (6.85 to 7.16)	-------------	0.671
Changes	−0.09 (−0.16 to −0.02)	0.26 (0.18 to 0.33)	0.34 (0.23 to 0.45)	<0.001 *
**Handgrip strength (kg)**				
Baseline	30.0 ± 4.3	30.7 ± 5.1	-------------	0.675
Changes	−0.52 (−1.60 to 0.53)	2.13 (0.91 to 3.39)	2.65 (0.75 to 4.56)	0.008 *
**Gait velocity (m/s)**				
Baseline	1.26 ± 0.15	1.25 ± 0.17	-------------	0.803
Changes	−0.03 (−0.05 to −0.01)	0.00 (−0.02 to 0.02)	0.02 (−0.06 to 0.01)	0.209 *

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
