# Peer review of "Effects of High Intensity Dynamic Resistance Exercise and Whey Protein Supplements on Osteosarcopenia in Older Men with Low Bone and Muscle Mass. Final Results of the Randomized Controlled FrOST Study"

_nutrients, 2020, doi:10.3390/nu12082341_

Round 1

Reviewer 1 Report

The present manuscript describes the results of a human study aimed at counteracting osteopenia and sarcopenia, considered as a unique entity called osteosarcopenia, by means of exercise training and protein supplementation.  The data are very interesting, deserve attention and are worthy to be shared with the scientific community. There are just few points that in my opinion require clarification:

  • Considering that in the excluding criteria, the lack of osteopenia or sarcopenia accounted for distinct individuals, there is a conflict with the initial statement about the unique osteosarcopenia entity. Please discuss.
  • It is hard to discriminate between HIT and WPS effect and whether one is determining most of the effect or both are equally needed, especially considering that also the CG group has a protein intake beyond the recommended values. Please expand the discussion on this point.

Minor points:

  • the excessive use of acronyms in the abstract renders the reading a bit difficult.
  • Figure legends should similarly describe all the acronyms.
  • Discussion: ‘exercise or more dedicated resistance exercise - ideally supported by dietary supplements - might be the only single agent’. Is it a single agent strategy?

Author Response

Dear Editors and reviewers

Thank you very much for your careful reviews of the article. We hope we have adequately answered all questions and addressed all suggestions and concerns. Changes and amendments in the text were highlighted and can thus be easily accepted or revised. Please find below the answers to the specific comments and the changes arising from the recommendations.

Reviewer 1

Open Review

English language and style

( ) Extensive editing of English language and style required
( ) Moderate English changes required
(x) English language and style are fine/minor spell check required
( ) I don't feel qualified to judge about the English language and style

A native British English speaking colleague again checked the manuscript.

Yes

Can be improved

Must be improved

Not applicable

Does the introduction provide sufficient background and include all relevant references?

(x)

( )

( )

( )

Is the research design appropriate?

(x)

( )

( )

( )

Are the methods adequately described?

(x)

( )

( )

( )

Are the results clearly presented?

(x)

( )

( )

( )

Are the conclusions supported by the results?

( )

(x)

( )

( )

We have revised the conclusion section.

the present manuscript describes the results of a human study aimed at counteracting osteopenia and sarcopenia, considered as a unique entity called osteosarcopenia, by means of exercise training and protein supplementation.  The data are very interesting, deserve attention and are worthy to be shared with the scientific community. There are just few points that in my opinion require clarification:

  • Considering that in the excluding criteria, the lack of osteopenia or sarcopenia accounted for distinct individuals, there is a conflict with the initial statement about the unique osteosarcopenia entity. Please discuss.

Sorry, this seems to be a misunderstanding: Of course, all participants suffer from osteosarcopenia. However, some men who have specific conditions (i.e. diseases, medications, life style factors) with confounding impact on our study endpoints were excluded. We have added corresponding information.

  • It is hard to discriminate between HIT and WPS effect and whether one is determining most of the effect or both are equally needed, especially considering that also the CG group has a protein intake beyond the recommended values. Please expand the discussion on this point.

We agree, we now discuss this issue in more detail. However, as stated, the study design did not allow us to assess the contribution of HIT-RT versus Protein supplementation. 

Minor points:

  • the excessive use of acronyms in the abstract renders the reading a bit difficult.

We agree, this is related to the limitation that only a total of about 200 words maximum were allowed.  We have now reduced the number of acronyms in the abstract

  • Figure legends should similarly describe all the acronyms.

We have revised the figure and now avoided acronyms. We further listed acronyms in the legends of the tables.

  • Discussion: ‘exercise or more dedicated resistance exercise - ideally supported by dietary supplements - might be the only single agent’. Is it a single agent strategy?

We agree – we have revised this sentence.

Reviewer 2 Report

To authors:

Authors examined an intervention study to determine the effects of resistance exercise on bone and muscle in community-dwelling older men. They found that resistance exercise with high intensity and supplementation of moderate whey protein, vitamin D, and calcium increased BMD at LS, maintained BMD at TH, and decreased risk for sarcopenia calculated by the sarcopenia Z score. The study is planned with an attention to the details, and the findings seem to be informative.

  • Why did they focus on men alone? Are any differences present in the effects of the HIT-DRT and WPS strategy on bone and muscle between women and men? Please discuss.
  • Another issue is regarding safety. Is there any deterioration of kidney function with supplementation of whey protein, vitamin D, and calcium in older men? Assessment by cystatin C clearance might be necessary to conclude it.

Author Response

Dear Editors and reviewers

Thank you very much for your careful reviews of the article. We hope we have adequately answered all questions and addressed all suggestions and concerns. Changes and amendments in the text were highlighted and can thus be easily accepted or revised. Please find below the answers to the specific comments and the changes arising from the recommendations.

Reviewer 2

Open Review

English language and style

( ) Extensive editing of English language and style required
( ) Moderate English changes required
(x) English language and style are fine/minor spell check required
( ) I don't feel qualified to judge about the English language and style

A native British English speaking colleague again checked the manuscript.

Yes

Can be improved

Must be improved

Not applicable

Does the introduction provide sufficient background and include all relevant references?

(x)

( )

( )

( )

Is the research design appropriate?

(x)

( )

( )

( )

Are the methods adequately described?

(x)

( )

( )

( )

Are the results clearly presented?

(x)

( )

( )

( )

Are the conclusions supported by the results?

(x)

( )

( )

( )

Comments and Suggestions for Authors

To authors:

Authors examined an intervention study to determine the effects of resistance exercise on bone and muscle in community-dwelling older men. They found that resistance exercise with high intensity and supplementation of moderate whey protein, vitamin D, and calcium increased BMD at LS, maintained BMD at TH, and decreased risk for sarcopenia calculated by the sarcopenia Z score. The study is planned with an attention to the details, and the findings seem to be informative.

  • Why did they focus on men alone? Are any differences present in the effects of the HIT-DRT and WPS strategy on bone and muscle between women and men? Please discuss.

The vast majority of exercise trials in the area of BMD/osteopenia/osteoporosis focus on postmenopausal women, therefore we decided to determine the effect in men. Further, while most exercise trials in women cohorts reported positive findings on BMD (review in [1]) favorable exercise effects in men are very rare [2]. Considering that comparable exercise protocols were applied, there might be indeed some evidence that the effects of HIT-DRT on BMD differ between men and women. In order to avoid a corresponding analysis on corresponding gender effects that might have decreased our statistical power further, we focus on the more challenging male cohort. 

  • Another issue is regarding safety. Is there any deterioration of kidney function with supplementation of whey protein, vitamin D, and calcium in older men? Assessment by cystatin C clearance might be necessary to conclude it.

Thank you for pointing this out. In the present FrOST study we conducted a comparable analysis of kidney and cardiac dysfunction/deteriorations than in the previous FranSO-study that  applied a combined high intensity (whole-body electromyostimulation) /whey protein supplementation (1.7-1.8 g/kg/d) approach in comparable men with sarcopenic obesity [3].So far, as in that trial, we have not detected relevant negative side effects in FrOST. Since we would like to publish these extensive data in a more dedicated contribution, we would like to exclude this issue in the present publication. However we have now addressed the point in the “limitation section” of the discussion.  

References

  1. Shojaa N, von Stengel S, Schoene D, Kohl M, Barone G, Bragonzoni L, et al. Effect of exercise training on bone mineral density in postmenopausal women: A systematic review and meta-analysis of intervention studies. Front Physiol. 2020;11:1427–1444.
  2. Kemmler W, Shojaa M, Kohl M, von Stengel S. Exercise effects on bone mineral density in older men: a systematic review with special emphasis on study interventions. Osteoporos Int. 2018.
  3. Kemmler W, von Stengel S, Kohl M, Rohleder N, Bertsch T, Sieber CC, et al. Safety of a Combined WB-EMS and High-Protein Diet Intervention in Sarcopenic Obese Elderly Men. Clin Interv Aging. 2020;15:953-967.

Reviewer 3 Report

The authors described an effective mean to improve osteosarcopenia using high-intensity, dynamic-resistance exercise training along with whey-protein supplementation in osteosarcopenic older men. This study is innovative and well-presented. It will provide a good method to improve osteosarcopenic pathology in older men.

Minor Points:

  1. Why were the participants not blinded for their exercise status?
  2. The authors need to provide details of diet records as this will affect the trial outcomes.
  3. Are there any record for incidence of fractures and changes in hormones in these patients that may be necessary to assess the stage of progression through the disease prior to their recruitment into the study. 

Author Response

Dear Editors and reviewers

Thank you very much for your careful reviews of the article. We hope we have adequately answered all questions and addressed all suggestions and concerns. Changes and amendments in the text were highlighted and can thus be easily accepted or revised. Please find below the answers to the specific comments and the changes arising from the recommendations.

Reviewer 3

Open Review

English language and style

( ) Extensive editing of English language and style required
( ) Moderate English changes required
(x) English language and style are fine/minor spell check required
( ) I don't feel qualified to judge about the English language and style

A native British English speaking colleague again checked the manuscript.

Yes

Can be improved

Must be improved

Not applicable

Does the introduction provide sufficient background and include all relevant references?

( )

( )

(x)

( )

Is the research design appropriate?

(x)

( )

( )

( )

Are the methods adequately described?

(x)

( )

( )

( )

Are the results clearly presented?

(x)

( )

( )

( )

Are the conclusions supported by the results?

(x)

( )

( )

( )

Comments and Suggestions for Authors

We have now introduced the rationale for the study in more detail

The authors described an effective mean to improve osteosarcopenia using high-intensity, dynamic-resistance exercise training along with whey-protein supplementation in osteosarcopenic older men. This study is innovative and well-presented. It will provide a good method to improve osteosarcopenic pathology in older men.

Minor Points:

  1. Why were the participants not blinded for their exercise status?

There are personal relationships among some participants that would have prevented a successful blinding strategy. Furthermore, the HIT-RT intervention was very prominent, thus an active control group with an exercise intervention that did not impact our study outcomes would have been easy to see through at least under the assumption that participants are aware of the intervention in the parallel group.     

  1. The authors need to provide details of diet records as this will affect the trial outcomes.

We have now added more details to this issue. However, we used a simple tool (Freiburger Ernährungsprotokoll), that focuses on the analysis of macro-nutrition.

  1. Are there any record for incidence of fractures and changes in hormones in these patients that may be necessary to assess the stage of progression through the disease prior to their recruitment into the study. 

We checked particularly the incidence of fragility fractures before and during the study. However, so far we have not analyzed our (frozen) blood samples for hormones, inflammatory markers or other interesting/important parameters. We are aware that this should not be an argument, but analyzing “state of the art” laboratory parameters is cost-intensive, even though our sample size is not very large. However, we are planning to determine at least some core parameters of muscle development and inflammation in close cooperation with colleagues who are more experienced in laboratory parameters.
